# A Comparison of Reflective Photoplethysmography for Detection of Heart Rate, Blood Oxygen Saturation, and Respiration Rate at Various Anatomical Locations

**DOI:** 10.3390/s19081874

**Published:** 2019-04-19

**Authors:** Sally K. Longmore, Gough Y. Lui, Ganesh Naik, Paul P. Breen, Bin Jalaludin, Gaetano D. Gargiulo

**Affiliations:** 1MARCS Institute for Brain, Behaviour and Development, Western Sydney University, Milperra, NSW 2560, Australia; g.lui@westernsydney.edu.au (G.Y.L.); ganesh.naik@westernsydney.edu.au (G.N.); p.breen@westernsydney.edu.au (P.P.B.); G.Gargiulo@westernsydney.edu.au (G.D.G.); 2Translational Health Research Institute, Western Sydney University, Campbelltown, NSW 2560, Australia; 3Ingham Institute of Applied Medical Research, University of New South Wales, Liverpool, NSW 2052, Australia; b.jalaludin@unsw.edu.au; 4Centre for Air Pollution, Energy and Health Research (CAR), Glebe, NSW 2037, Australia

**Keywords:** photoplethysmography, heart rate, SpO_2_, respiration rate, anatomical location

## Abstract

Monitoring of vital signs is critical for patient triage and management. Principal assessments of patient conditions include respiratory rate heart/pulse rate and blood oxygen saturation. However, these assessments are usually carried out with multiple sensors placed in different body locations. The aim of this paper is to identify a single location on the human anatomy whereby a single 1 cm × 1 cm non-invasive sensor could simultaneously measure heart rate (HR), blood oxygen saturation (SpO_2_), and respiration rate (RR), at rest and while walking. To evaluate the best anatomical location, we analytically compared eight anatomical locations for photoplethysmography (PPG) sensors simultaneously acquired by a single microprocessor at rest and while walking, with a comparison to a commercial pulse oximeter and respiration rate ground truth. Our results show that the forehead produced the most accurate results for HR and SpO_2_ both at rest and walking, however, it had poor RR results. The finger recorded similar results for HR and SpO_2_, however, it had more accurate RR results. Overall, we found the finger to be the best location for measurement of all three parameters at rest; however, no site was identified as capable of measuring all parameters while walking.

## 1. Introduction

Photoplethysmography (PPG) is the non-invasive use of light interactions with the optical properties of tissue, blood vessels, and blood, often used to estimate physical parameters of the cardio-vascular and respiratory system [1]. The use of PPG as a non-invasive method to detect heart rate was demonstrated by Alrick Hertzman [2,3] in 1938, using a pencil flashlight and a photoelectric cell connected to an electrocardiograph, to record the cardiac cycle through pulsative engorgement and dis-engorgement of blood vessels. PPG sensors today use the same principle as the device introduced by Hertzman, however, these have been miniaturised using modern light emitting diodes (LEDs), photodiodes, and microcontrollers [1].

PPG devices are in common use throughout the healthcare system, providing a quick non-invasive method for obtaining heart rate and blood oxygen saturation (SpO_2_) in a clinical setting. Most often in these settings transmissive PPG devices are utilised on the patient’s finger, however, sensors designed for the toe, nasal septum, earlobe, and forehead are also utilised where a finger-based sensor is not feasible [4,5]. In recent years, PPG sensors have been integrated into wearable devices designed to be worn on the user’s wrist, such as the Apple Watch and Fitbit devices, for the monitoring of heart rate during exercise and other daily activities [6]. Most of these devices make use of reflective PPG using the green and/or infrared wavelengths, and therefore are generally incapable of measuring SpO_2_, as detecting differences in light absorption of oxyhaemoglobin and de-oxyhaemoglobin requires light in the red and infrared part of the spectrum [7].

Heart rate, respiration, and SpO_2_ are three of the five vital signs indicating the criticality condition of a patient (aside from blood pressure and temperature). If any of these vital signs have deteriorated, it is an indicator that the patient is in immediate need of medical intervention [8,9,10]. With the introduction of PPG based pulse oximeters, heart rate and SpO_2_ can easily be monitored using a single device, and respiration is often monitored visually or through separate devices as required. A single device which can monitor all three signs is a desirable diagnostic tool for emergency medicine, as well as for the monitoring and diagnosis of other diseases, such as sleep apnoea [8,9,10].

While many studies have been previously conducted into the placement of PPG sensors for pulse oximetry [5,10,11,12,13,14,15,16,17], few studies have been conducted for a suitable location to collect heart rate, SpO_2_, and respiration. Kramer et al. [10] developed a new pulse oximeter for use at non-traditional anatomical locations for the measurement of heart rate, SpO_2_, and respiration. They tested their device at five anatomical locations (pectoral, sternum, bicep, calf, and forearm) and found that the calf location recorded the most accurate SpO_2_ measurement [10]. Despite respiration artefacts, Kramer et al. [10] found that the sternum and pectoral locations demonstrated acceptable SpO_2_ accuracy, while the forearm and bicep performance was significantly degraded compared to other tested locations. It was also shown that the respiration rate could be captured with the sensor located on the pectoral, the only location where the authors attempted respiration data extraction from the PPG waveform [10]. Ground truth for respiration was obtained from 10 of the 42 participants by estimating respiration in a capnography waveform, limiting the significance of the results from the remaining 32 participants. SpO_2_ was measured against baseline SaO_2_ collected from arterial blood drawn throughout the experiment. Estimation of heart rate accuracy was excluded from the study as a suitable ground truth for heart rate was not included [10].

Nilsson et al. [18] measured heart rate and respiration using a PPG pulse oximeter at five anatomical locations: The finger, forearm, wrist, shoulder, and forehead. Three different wavelengths of light (green, red, and infrared) were assessed, as well as reflective and transmissive PPG. The paper found that the heart rate was detectable at all locations, emission wavelengths, and methods of PPG tested. The respiration waveform was detectable at each location, with varying degrees of accuracy. Respiratory rate accuracy was greatest at the forearm, while the finger produced the most accurate heart rate measurement. The forehead location had a good balance of accuracy between heart rate and respiration. The author concluded that sites where good respiration signal was present, generally had poorer heart rate signal, and the opposite was true for sites where the respiration signal was not as strong. One of the issues with this study was that the sensors at each location were different and different wavelengths were used at different locations, which may have impacted on the results of the research as indicated in Table 1. Additionally, the study did not investigate the ability to obtain SpO_2_ at any of the locations, one of the key parameters of our study [18].

The aim of this study is to evaluate the use of red and infrared reflective PPG for pulse oximetry and respiration rate extraction at eight anatomical locations for pulse oximetry and respiration, at rest and during walking. We hypothesised that heart rate and SpO_2_ would be more accurate at positions where shallow dermal tissue is present, and the contact between the sensor and the skin experiences minimal movement. It was further hypothesised that the respiration rate (breaths per minute) signal will be strongest when the sensor was located at lower extremities due to hydrostatic pressure exerted by the diaphragm on the lower cardiovascular system during respiration. Our overall aim was to establish an optimal position on the body whereby a reflective PPG sensor occupying approximately 1 cm^2^ of skin can accurately detect and record the three most important bio-parameters, heart rate, SpO_2_, and respiration, at rest and during walking.

## 2. Materials and Methods 

The materials and methods are divided into sections covering the hardware utilised for sensing, associated firmware, the methods employed for data collection, and analysis of the data.

### 2.1. Sensor Hardware

The sensor hardware consisted of eight Maxim Integrated™ MAXREFDES117 development boards featuring the red/infrared MAX30102 PPG sensor (Maxim Integrated™, San Jose, CA, USA) mated to a NodeMCU Amica (Expressif Systems ESP8266-based) microcontroller board through an Adafruit® breakout board utilising the Texas Instruments™ TCA9548A 1 to 8 I^2^C multiplexer (Texas Instruments™, Dallas, TX, USA) as detailed in Figure 1. A power distribution rail was designed using 4 capacitors to ensure stable 3.3 V DC power was available to all attached devices (Figure 2). The NodeMCU microcontroller was connected to a Windows 10 PC via a USB cable to the NodeMCU’s integrated serial interface via a USB powered hub with a 1A power supply, supplying sufficient power to the sensors, multiplexer, and microcontroller. The microcontroller, power rail and I^2^C multiplexer were housed in a 3D printed enclosure (Figure 2).

The MAXREFDES117 development board contains a MAX30102 PPG sensor at its heart and supporting components. The MAX30102 PPG sensor contains two light emitting diodes (LEDs), one infrared (peak wavelength 880 nm) and the other red (peak wavelength 660 nm); along with a photodiode that is specific for wavelengths between 600 and 900 nm. Included on the MAX30102 sensor is an integrated temperature sensor for measuring temperature variations, an analogue to digital converter (ADC), and an I^2^C interface. By integrating the ADC and I^2^C interface into the sensor itself, noise and artefacts generated between the photodiode and the ADC are minimised. Furthermore, the values measured by the sensor are transmitted with a checksum, such that the validity of the data is verified when it is received by the supporting microcontroller.

Firmware for the microcontroller was developed such that the I^2^C multiplexer was switched through each I^2^C channel sequentially, reading the value of each PPG sensor read and write pointer. A mismatch between the sensor write pointer and read pointer values indicates new PPG data are available in the buffer. The firmware read all available data in the buffer until the read and write pointers were again equal, before moving on to retrieve the next sensor’s read and write pointer values. Data was sent immediately via serial connection to the connected Windows 10 PC, where the data was piped to a file via a serial monitor for later channel extraction and analysis. The PPG sensor boards were configured to the sample rate of 100 Hz using the settings set out in Table 2. Firmware is available upon request to the author.

Six of the eight MAXREFDES117 development boards were housed in a 3D printed enclosure providing an airgap of approximately 2 mm between the sensor window and the skin (Figure 3). The forehead sensor was sewn into the brim of a hat such that the sensor window would rest directly on the skin of the forehead (Figure 3). Finally, the finger sensor was placed within a commercial pulse oximeter clamp housing, replacing the commercial pulse oximeter electronics, to provide the best approximation to a commercial fingertip sensor (Figure 3).

Respiration ground truth data was collected via three analogue electro-resistive bands fitted within the respiration shirt, connected via an analogue to digital converter (ADInstruments PowerLab 8/35) connected to a PC, using the methods and shirt developed by Gargiulo et al. [19,20]. The respiration shirt was fitted with multiple electro-resistive bands, however, only data from the chest band was used in this study (Figure 4). The analogue to digital converter was connected to a Windows 10 PC via a USB running LabChart 8, and data was recorded at 1 kHz, and saved to file in the MATLAB data format.

Ground truth heart rate and SpO_2_ data was collected manually at 15 s intervals from a commercial finger based transmissive PPG pulse oximeter (CONTEC™ CMS50D Pulse oximeter, Contec Medical Systems, Qinhuangdao, China) worn on the finger, and entered into a Microsoft Excel file for data analysis.

### 2.2. Data Collection Method

A PPG sensor was placed on the second digit of the right hand, the forehead via a cap, the right tibia supported via Velcro attached to a sports brace, the right wrist (posterior distal forearm), the rib cage near the fifth and sixth ribs on the right hand side, the neck at approximately the C7 vertebrae, the lower back at approximately the L5 vertebrae, and the temple region to the right of the right eye (Figure 4). All PPG sensors other than the forehead and tibia were taped in location within their respective 3D printed housings using sports tape. The commercial pulse oximeter was located on the second digit of the right hand (Figure 4). The respiration shirt was worn over the top of the sensors placed on the back, neck, and ribcage, and tightened using the supplied straps so that it retained a snug fit and respiration was clearly visible in LabChart 8 (Figure 4).

Six data sets were collected while resting seated in a chair from four of the authors (one female, three males). A further six data sets were collected from all subjects while walking for seven minutes at a pace of 4 km/h on a treadmill. Each sample set consisted of seven minutes of data collection from all eight PPG sensors, the manual data collected from the commercial pulse oximeter, and the respiration shirt. Data collection was initiated from each system (PPG sensors, respiration shirt, and manual recording) as close as possible to each other, however, a small synchronization error was deemed inconsequential as the manual data collection had the lowest resolution at 15 s for heart rate and SpO_2_, while respiration data for the shirt and as calculated from the PPG data was averaged over 60 s.

Manual data was collected from each subject by visually reading the SpO_2_ and HR values on the commercial pulse oximeter at 15 s intervals as recorded on a stop watch. These values were noted on paper and transferred to a Microsoft Excel spreadsheet, before being saved in CSV format for data processing. PPG sensor data was initiated by connecting via a serial monitor to the microcontroller and piping the output to a file. LabChart 8 was used to record the respiration shirt data. At the end of recording a dataset, the recorded respiration shirt data was saved in the LabChart 8 format and MATLAB workspace format.

### 2.3. Data Processing and Analysis

A script was written to extract the data from the PPG output file and the manual commercial sensor collection, and written into a single CSV output file. Linear interpolation was used to up sample the manual dataset, enabling comparison of the test PPG heart rate and SpO_2_ data at 100 Hz with a manual data set recorded at 15 s intervals.

A second script was written in MATLAB R2018b to import the PPG and manual data from the resultant CSV file and import the respiration shirt MATLAB datafile and then perform data processing for analysis. The first step in processing the PPG data was to pass the data through a 50th order IIR filter that includes a bench of notches to remove powerline interference and on each sensor channel (red and infrared) for each sensor. The data was then passed through a filter to remove the DC component of the data, and finally passed through the low pass filter with a 3 Hz cut-off.

Next, the peak and valley locations and values were located using MATLAB’s built-in *findpeaks* function. This function, however, does not always find the actual peaks and valleys, but often a nearby location. The actual peaks and valleys were found using max/min functions searching 5 samples either side of the locations found using the *findpeaks* function.

The instantaneous heart rate was calculated for each sensor and dataset using the time between peaks, with the timestamp for heart rate calculated as the mean of the time of the two peaks. The heart rate was then averaged over the time between timestamps, interpolating the instantaneous heart rate such that an estimate of the instantaneous heart rate at any 10 ms timepoint in each dataset was formed. We then calculated the average heart rate at 5, 15, and 60 s, and calculated the percent difference compared to the estimated manually recorded heart rate from the commercial sensor.

Blood oxygen saturation (SpO_2_) was calculated using a formula supplied in the Maxim Integrated™ sample code [21]. First, the AC and DC component of the pulsative waveform was calculated for both the red and infrared channels and stored in integer variables (*AC_Red_, AC_IR_, DC_Red_, DC_IR_*) as a mean of 5 consecutive peaks/valleys. A ratio (*R*) of the AC and DC was then calculated from the mean AC and DC values (Equation (1)), and the SpO_2_ value was calculated using the Maxim Integrated™ formula as shown below (Equation (2) [21]):(1)$$R=\;\frac{AC_{Red}÷DC_{Red}}{AC_{IR}÷DC_{IR}}$$
(2)$$SpO_{2}=-45.060\times R^{2}+30.354\times R+94.845$$

The timestamp for the calculated SpO_2_ value was derived as the mean of the timestamps for the first value and last value of the 5 maximum values used to derive the *AC_Red_* value. The resultant values were then interpolated using the same method as per the heart rate such that an estimate of the SpO_2_ value at each 10 ms interval was generated. Using the interpolated dataset, SpO_2_ was calculated at 5, 15, and 60 s intervals, and compared to the manually recorded SpO_2_ values.

Respiration ground truth was then extracted from the respiration shirt dataset using MATLAB. The raw chest shirt data and PPG data for each location and channel were then filtered using a low pass filter with a 0.1 Hz cut-off frequency and a Kaiser window with a beta of 3. The data was then passed through a Butterworth filter with a filter order of 4 and a cut-off frequency calculated as follows, where the respiration shirt had a 1000 Hz sample rate and the PPG data had a sample rate of 100 Hz.

The respiration rate was then calculated by initially splitting the datasets up into several windows of a defined window size (15, 30, and 60 s), before searching for the peaks and the valleys in MATLAB to obtain the average peak/valley prominence. Using the average peak/valley prominence, another search for peaks and valleys was conducted to obtain the location of each peak within each defined window of data. The instantaneous time between peaks was then calculated for the window, with outliers removed using the MATLAB *rmoutlier* function. The resultant list of instantaneous respiration rates was then averaged over the defined window to produce an array of average values.

Finally, for all heart rate, SpO_2_, and respiration 60 s average datasets, the error was calculated compared to the expected value from the commercial pulse oximeter and the respiration shirt, and expressed as a percent error using the following formula (Equation (3)):(3)$$error\%=\;\frac{\left|expected\;value-actual\;value\right|}{expected\;value}\;\times \;100$$

### 2.4. Ethics Statement

Participants of this preliminary study consisted of the authors and therefore ethical approval was not deemed necessary. Ethical approval will be sought for an extension of this study to a statistically significant population of volunteer participants in the next stage of this study.

## 3. Results

The participants of this preliminary study consisted of four of the paper’s authors, one female (42 years of age) and three males (29, 41, and 45 years of age). The raw PPG data was processed according to the protocol detailed in the methods section. Data for heart rate, SpO_2_, and respiration rate were calculated using 15, 30, and 60 s windows on each dataset collected (7 min). Additionally, the data was calculated per PPG channel (red, IR) for heart rate and respiration, as well as calculation of an average for both channels of data (both). Finally, the data for heart rate, SpO_2_, and respiration was calculated for the studies conducted at rest (rest) and while walking (walk) at 4 km/h. Percentage error for all three parameters was calculated compared to their respective ground truth data for the window each dataset was calculated in (15, 30, and 60 s). No datapoints were dropped from the study.

PPG waveforms were successfully processed at the finger and forehead to reveal the heart rate signal for peak to peak detection of heart rate (Figure 5), and the amplitude of each beat required for SpO_2_ calculation compared to the unprocessed waveform (Figure 5 and Figure 6). However, the filters applied were less successful in producing defined peaks and valleys associated with blood vessel volume and blood oxygen composition compared to other measurement locations in this study, as shown by the lower back and rib cage shown in Figure 6. Additionally, oscillations in amplitude of the processed waveform due to respiration were clearly present in the finger and forehead PPG signal (Figure 5 and Figure 6).

### 3.1. Heart Rate

At rest, the finger was found to have the highest accuracy, with a median error of 1.5% (±16%) using both the red and IR channels combined; however, the forehead recorded a similar median error, but with a smaller standard deviation (1.4% ± 8.3%) (Table 3). In the walking dataset, the forehead (4.3% ± 7.6%) was more accurate than the finger (6.5% ± 10%) when comparing the combined red and IR data (Table 3). When comparing the red channel alone, the finger (1.4% ± 18%) was more accurate than the forehead (1.5% ± 17%) at rest, while the reverse was true while walking (finger 6.8% ± 11%, forehead 4.6% ± 7.8%) (Table 3). The IR channel for the forehead (rest 0.68% ± 1.5%, walk 4.2% ± 7.5%) was more accurate while both resting and walking, compared to the finger (rest 1.2% ± 16%, walk 5.9% ± 10%) (Table 3).

Visually comparing the waveforms between resting and walking (Figure 7), the finger and forehead both contain a strong heart rate signal visible in both the red and IR waveform during the resting and walking tests. The other locations tested contain waveforms with no evident blood vessel volume while resting, and contain evidence of movement artefacts while walking that do not compare to the peaks visible in the finger and forehead waveforms (Figure 7).

### 3.2. SpO_2_

As was found with the heart rate results at rest, the forehead (2.0% ± 1.1%) and finger (2.1% ± 1.2%) were most accurate for SpO_2_ data (Table 4). The temple (2.7% ± 7.7 × 10^4^%) had a slightly higher median error than the finger with a higher standard deviation, while the neck (8.4% ± 47%) was the fourth most accurate location (Table 4). All other locations at rest recorded a median error larger than 20% (Table 4). During the walking test, the finger (2.2% ± 6.3%) was most accurate with the forehead second (2.8 ± 2.2%); however, the finger also had a three times higher standard deviation than the finger. The temple (6.7 ± 5.9 × 10^2^%) was third most accurate during the walking test, but as with resting, the temple had a high standard deviation. All other locations recorded a median error greater than 19% during the walking test (Table 4).

In a visual comparison of the infrared and IR waveforms for the finger and forehead (Figure 7), a noticeable amplitude difference is evident between the red and IR waveforms, indicating information on the blood composition with regard to oxyhaemoglobin and de-oxyhaemoglobin is present for the calculation of SpO_2_. For all other locations tested, there is negligible difference between the red and IR waveforms while resting and walking, indicating there is insufficient information present for which to calculate SpO_2_ (Figure 7).

### 3.3. Respiration

Respiration waveforms were successfully processed as set out in the material and methods section. The waveforms for the finger, lower back, and rib cage show close synchronization with the baseline waveform collected from the respiration shirt data, and the waveform for the forehead did not have a respiration waveform present (Figure 8). The waveform for the finger is less apparent during walking. The waveform for the lower back and rib cage contains movement artefacts, and the forehead does not appear to have a respiration waveform present (Figure 8).

At rest, the rib cage recorded the lowest median error for respiration in both the red (0.13 respirations per minute (rpm) ± 8.7) and infrared channels (0.19 rpm ± 6.8) (Table 5). However, all other locations except the forehead and temple recorded a median error less than 1.1 rpm (Table 5). While walking, the lowest median error was recorded on the rib cage in the infrared (0.36 rpm ± 13), with the rib cage in red the second most accurate (2.9 rpm ± 13). All other locations recorded errors above 3.9 rpm (Table 5).

### 3.4. Overall

Overall, the locations with the lowest median error was the finger, followed by the forehead and temple (Figure 9), with the same ranking while walking and resting. The wrist displayed the highest error overall, followed by the tibia and rib cage, although the error for the wrist while walking was lower than the lower back and tibia (Figure 9). The overall results reveal the finger as the best location for recording heart rate, SpO_2_, and respiration data simultaneously using PPG, with the forehead second (Figure 9).

## 4. Discussion

### 4.1. Single Site Heart Rate, SpO_2_, and Respiration Rate

The primary goal of this study was to determine a single site whereby a 10 mm × 10 mm PPG sensor could obtain the heart rate, SpO_2_, and respiration. In our study, we found that obtaining heart rate and SpO_2_ within an acceptable median error and standard deviation while both at rest and walking was only possible at a single tested anatomical location, the forehead. Despite the infrared PPG waveform clearly containing a respiration waveform on some tests, as evidenced by Figure 5 and Figure 6, the forehead was unable to demonstrate an acceptable accuracy for measurement of respiration during rest or while walking overall. It has been demonstrated in previous studies that light penetrates skin to a shallower depth in subjects with a darker skin tone. The test cohort in this study consisted of a wide range of skin tones, therefore, it is possible that the respiration accuracy on the forehead may have been affected by darker shades of skin.

We also found that obtaining heart rate, SpO_2_, and respiration within an acceptable degree of accuracy was also possible at a single tested anatomical location at rest, the finger. The use of both reflective and transmissive PPG for pulse oximetry on the finger is well accepted and has been demonstrated in numerous publications [12,22,23]. The results in this study for the finger pulse oximetry at rest concur with the literature. However, while walking, the finger recorded a significant increase in median error for SpO_2_ and respiration, such that this location was not suitable for the measurement of these values while walking. The higher SpO_2_ and respiration on the finger while walking was likely due to movement artefacts introduced into the waveform from each step. It is possible that the error can be reduced through additional filtering or by using accelerometer data to dampen the interference as each step is taken.

The accuracy of respiration data extraction at the fingertip at rest was expected, and consistent with the published literature. In a study conducted by Addison et al. [24], using a commercial pulse oximeter affixed to the finger, a root mean squared error (RMSE) of 1.83 breaths per minute was obtained, which corresponds with the results of our study. Addison et al. [24] did not measure the accuracy of the heart rate or SpO_2_, however, as they utilised a commercial hospital grade PPG pulse oximeter, and it can be assumed that the sensor was capable of obtaining those parameters. While Addison et al. [24] utilised unguided breathing in their study, they did not study the effectiveness of the device during exercise, a key difference with the present study.

Nilsson et al. [18] compared multiple anatomical sites for their ability to measure respiration and heart rate. Nilsson et al. [18] found that of the five measurement locations tested (forearm, finger, forehead, wrist, and shoulder), none of the sites were identified as an obvious location for the detection of both breathing and heart rate, with both signals detected at all tested locations. While our study only identified the finger at rest as being suitable for the detection of respiration, heart rate, and SpO_2_, the other sites tested either performed poorly for respiration and well for pulse oximetry, or well for respiration and poorly for pulse oximetry. Therefore, even if we only include respiration and heart rate as desired parameters for a single-site sensor, our results do not indicate any sites other than the finger where respiration and heart rate could reliably be detected, a result which is not consistent with the study by Nilsson et al. [18]. This points to a confounding variable in our study, which was not identified at the onset of the experiment. However, one consistency between our findings and those of Nilsson et al. [18] is that both studies found that where the respiration signal was the strongest, the heart rate signal was often the poorest, while the opposite was true where the heart rate signal was the strongest.

### 4.2. Pulse Oximetry Site Comparison

One of the secondary goals of this study was to determine anatomical locations whereby pulse oximetry can be measured accurately, with minimal intrusiveness to the user over a long period during normal day to day activities. With the exception of the finger, the locations chosen in our study correspond to locations where a pulse oximeter may be worn over extended periods of time. Our results revealed that the forehead is the best location for the measurement of pulse oximetry during normal day to day activity, with an overall median error of 2.9% for heart rate (red/infrared resting 1.4% ± 8.3%, walking 4.3% ± 7.6%), and 2.4% for SpO_2_ (resting 2.0% ± 1.1%, walking 2.8% ± 2.2%) compared to manually recorded data from the commercial finger-based pulse oximeter. With the forehead sensor mounted on the brim of a hat, the forehead makes an ideal place for measurement of pulse oximetry during day to day activities, especially in outdoor environments where a hat is recommended for sun protection or for work safety requirements. As discussed earlier, the finger (resting 2.1% ± 1.2%, walking 2.2% ± 6.3%) also produced acceptable results for pulse oximetry, however, the finger displayed a higher standard deviation than the forehead during the walking tests.

All other sites tested in the study were unable to record acceptable heart rate and SpO_2_ data combined. Interestingly, for all sites tested, excluding the finger and forehead, the accuracy for heart rate detection increased during the walking tests compared to the resting tests. Similarly, issues were found with SpO_2_ measurements at all sites except for the finger and forehead, with the median error for the rib cage, lower back, and tibia over 20% at rest, and all sites except the forehead, finger, and temple measuring over 19% median error while walking. The difference between the red and infrared channels for heart rate measurement was not statistically significant at all test sites.

Although it was expected that the accuracy for PPG heart rate and SpO_2_ measurement at the non-traditional anatomical locations would be degraded compared to the more traditional locations of the forehead and finger, it was not expected that the error would be as large as what was recorded in this study. One of the possible reasons for this is the enclosure used throughout this study. At the traditional locations of the forehead and finger, there was little to no air gap between the sensor and the finger. The PPG sensor located on the forehead was mounted into the brim of a hat, with the sensor window placed directly against the skin, while the sensor for the finger was placed in a modified commercial finger clip PPG sensor housing, with a <1 mm airgap between the sensor window and the skin, while the 3D printed sensor housing for the other sensor locations possessed a ~2 mm airgap between the sensor window and the skin. This airgap in the 3D printed sensor housing may have contributed to the inaccuracy of pulse oximetry recorded at non-traditional locations by altering the photon path designed into the PPG sensor, rendering the heart rate and oxy/deoxy-haemoglobin component of the waveform unreadable. Additionally, the 3D printed sensor housing was printed in a white filament, which may have allowed excessive ambient light to flood into the airgap or additional scattering of light, further masking the heart rate and blood composition component of the waveform. 

The study by Nilsson et al. [18] researched the placement of PPG sensors for heart rate and respiration, whereby we discussed how the finger was suitable for heart rate and respiration. However, the other sites tested by Nilsson et al. [18], i.e., the forearm, forehead, wrist, and shoulder, were capable of determining heart rate within an acceptable margin of error. As with our results, the finger and forehead both produced a good heart rate signal in the Nilsson et al. [18] study. Unlike our study, Nilsson et al. [18] found that the wrist produced acceptable results for heart rate calculation, whereas we were unable to detect the heart rate at the wrist to within an acceptable degree of error. Nilsson et al. [18] did not test for SpO_2_.

In a study investigating the ability to monitor SpO_2_ at different measuring locations, Kiruthiga et al. [13] tested four locations, the wrist, forehead, finger, and chest. It was found that the finger provided the best location for the determination of SpO_2_ in the test subjects, followed by the forehead. These results agree with the results of our study, whereby the forehead and finger were found to be most accurate for SpO_2_ at rest; however, we found the forehead was more accurate by a small degree. Additionally, Kiruthiga et al. [13] found that SpO_2_ was detectible on the wrist, but depended on the sensor placement (SpO_2_ was detectable when the sensor was placed closer to the distal and ulnar arteries) and was susceptible to motion artefact. The chest displayed the highest margin of error. Unlike this study, we were unable to detect SpO_2_ on the wrist or the rib cage (chest).

### 4.3. Respiration Site Comparison

Extraction of respiration information from the PPG waveform at various sites was another goal of this study. Only the rib cage was capable of recording respiration with a median error below 0.5 rpm at rest and also while walking. At this location, we demonstrated a median error of 0.19 rpm (±6.8) using the infrared channel at rest, and 0.36 rpm (±13) while walking (Figure 6). While the red channel at the chest had the lowest median error at rest (0.13 rpm ± 6.6), walking increased the median error to 2.9 rpm (±13) (Figure 6). As opposed to all other locations tested for respiration, the rib cage was unique in that it did not measure respiratory induced intensity variation (RIIV) in the blood volume during breathing, but rather movement of the sensor during expansion and contraction of the rib cage during each breath. This was apparent from the easily visible respiration cycle that could be seen in the raw unprocessed waveform from the PPG sensor at the rib cage location. However, even though the waveform contained a strong respiration signal at rest, this signal was masked from the steps taken while walking at 4 km/h, resulting in decreased accuracy.

As discussed earlier, respiration extraction from fingertip PPG was demonstrated at rest in both the red (1.1 rpm ± 7.7) and infrared channels (0.2 rpm ± 7.7), consistent with the existing literature, however, it was not possible to obtain respiration while walking using either wavelength at this location. Today, commercial PPG pulse oximeters are available that are capable of measuring heart rate, SpO_2_, and respiration, such as the Masimo MightSat™ Fingertip Pulse oximeter at rest.

At rest, all locations below the head recorded respiration with a median error below 1.1 rpm at rest, demonstrating that each of these locations were suitable for respiration data collection at rest. There was little difference at rest between the red and infrared channel at these locations. The neck in the red channel had a median error of 0.26 rpm (±8.7) with an increased median error of 1.1 rpm (±8.7) in the infrared. Conversely, the wrist was worse in the red channel with a median error of 1.1 rpm (±7.7), compared to the infrared channel at 0.2 rpm (±7.7). Interestingly, apart from the rib cage discussed earlier, the tibia displayed the highest accuracy in both the red and infrared channels with a median error of 0.29 rpm (±8.3) for the red and 0.21 rpm (±8.4) for the infrared. Comparatively, locations on the head recorded a high degree of error for respiration extraction, with the forehead recording a median error of 6.6 rpm (±16) and the temple recording 4.5 rpm (±10) in the red channel at rest, with both locations recorded a higher error in the IR channel at rest (forehead 18 rpm ± 21, temple 5.4 rpm ± 12). All locations recorded an increased mean error during the walking tests compared to the results at rest. At the tibia, for example, the error increased from an acceptable value at rest in both the red (0.29 rpm ± 8.3) and IR channels (0.21 rpm ± 8.4) to over 20 times larger while walking (red 26 rpm ± 11, infrared 19 rpm ± 12). The large error during exercise compared to the acceptable value at rest is likely due to excess movement of the sensor while walking, in addition to muscular movement of the leg. The only location where the results were unaffected by walking, compared to the results at rest, was on the forehead in infrared, however, this location did not demonstrate an acceptable level of accuracy at rest or walking.

While our results for forehead respiration measurements indicated that this location is not an ideal site for respiration extraction, this is not consistent with the literature. It should be noted, however, that the respiration waveform is clearly visible in some datasets, as evidenced by Figure 6 and Figure 7, therefore, respiration data can be obtained at this location. One possible reason for the failure to extract respiration in this study may be that the algorithm used was a peak detection algorithm using the respiratory induced intensity variation (RIIV) component of the PPG waveform. The RIIV waveform is induced by intra-thoracic pressure changes during inhalation and exhalation, resulting in a change over time in the baseline of the PPG waveform [18,25,26,27,28,29,30]. However, there are other techniques which can be used to extract the respiration waveform, including the respiratory induced frequency variation (RIFV), in which the autonomic nervous system regulates the heart rate in response to the respiratory cycle, and the respiratory induced amplitude variation (RIAV) described as a variation in the strength of the heart rate during inspiration caused by the filling of the ventricular chamber of the heart [25,30,31]. During future experiments, these alternative methods of respiration extraction may yield reasonable respiration measurement on the forehead and at other locations.

Experimentally, other publications have demonstrated that the forehead is a viable location for respiration extraction. Although van Gastel et al. [25] used a contactless method for PPG respiration measurement, which differs from our skin contact PPG approach, their study demonstrated that the forehead was better for respiration detection than the finger, which contradicts the results from our study. Besides the method of PPG waveform recording, the other major difference was that van Gastel et al. [25] utilised the IR part of the spectrum, as well as the full visible spectrum for PPG waveform recording, whereas we utilised only the red and IR part of the spectrum. The visible spectrum in their study yielded a better signal to noise ratio compared to the IR channel. Therefore, the use of a wider part of the visible spectrum through the use of a three wavelength PPG sensor (green, red, and IR) on the forehead and at other locations may provide an improved respiration signal than the the bi-colour sensor used in this paper.

As discussed earlier, in the study conducted by Nilsson et al. [18], multiple anatomical sites were tested for their ability to measure respiration and heart rate. With regards to respiration, Nilsson et al. [18] found that of the five measurement locations tested (forearm, finger, forehead, wrist, and shoulder), the respiration signal could be identified and extracted successfully at each tested location. Despite the successful extraction of the respiration signal from each site, there was a marked difference in the accuracy of the respiration signal coherence at each tested location. The forearm in the Nilsson et al. [18] study produced the strongest coherence to the reference respiration signal, followed by the shoulder and forehead. The wrist and finger displayed the lowest coherence with the reference signal. These results are not consistent with our findings, whereby similar locations to those used by Nilsson et al. [18], the finger and wrist, produced the lowest respiration mean error, while the respiration signal at the forehead was essentially undetectable. These inconsistencies are another indication that a confounding variable was missed in our study, likely a result of the air gap in the 3D printed enclosure or the transparency of the enclosure introducing outside light source interference.

### 4.4. Limitations

As a preliminary study into single site minimally invasive heart rate, blood oxygen saturation, and respiration measurement, the primary limitation in this study is the number of participants, which reduces the significance of the results. However, we intend to extend this study to a larger gender balanced population in the future. Additionally, while the use of a respiration shirt containing resistive bands has been previously compared against a spirometer by Gargiulo et al. [20], the respiration shirt was not tested during exercise. Additionally, the respiration shirt is not the standard for respiration measurement, therefore, future testing will utilize a spirometer or a pneumotachometer for respiration ground truth. As discussed earlier, the housing used for PPG devices at all anatomical locations may have interfered with the sensors’ ability to detect changes in blood volume associated with the cardiac cycle and blood oxygen composition. To alleviate the sensor housing’s excessive air gap and transparency concerns, a new opaque flexible skin-safe silicone housing will be developed for the PPG sensors, providing a housing that can adhere to the skin, minimising air gaps, as well as preventing light from corrupting the PPG signal. Finally, our baseline measurement for pulse oximetry was manually recorded at 15 s intervals from a commercial fingertip pulse oximeter, introducing synchronisation errors between the baseline and the experimental data. In our future study, a Bluetooth-based pulse oximeter will be used such that baseline pulse oximetry data can be recorded continuously throughout the experiment, and a synchronous initiation of data capture can be achieved.

## 5. Conclusions

In this study, we demonstrated that the finger was the best location for simultaneous measurement of HR, SpO_2_, and respiration rate while at rest and walking. However, we showed that the forehead is more suitable if only HR and SpO_2_ measurements are required. With the sensor on the forehead contained within the brim of a hat, this location frees up the patient to continue daily tasks with minimal interference, while continuously recording HR and SpO_2_. With more research and more capable PPG sensors, it is envisaged that the respiration rate can be obtained on the forehead, rendering the forehead as the ideal location for HR, SpO_2_, and respiration rate data collection, with minimal interference to day to day activities.

## Figures and Tables

**Figure 1 sensors-19-01874-f001:**
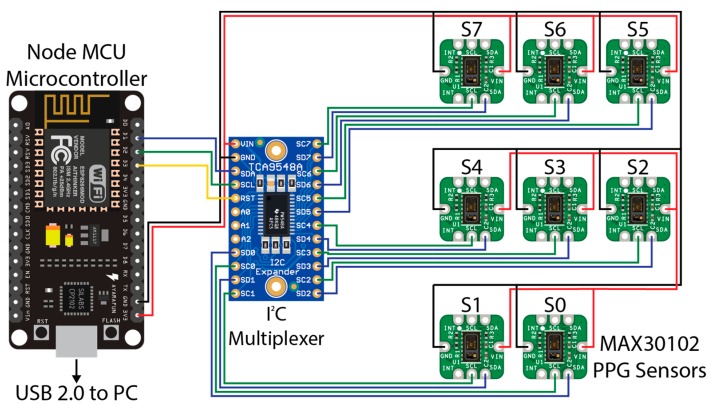
Wiring diagram for the PPG sensors used in this experiment. A NodeMCU microcontroller (left) was connected to a PC (not shown) via USB 2.0 for data recording. A TCA9548A I^2^C multiplexer (centre) was connected to the NodeMCU microcontroller via pins D1 (SDA/blue wire), D2 (SCL/Green Wire), and D3 (RST/Yellow Wire). Eight MAX30102 PPG sensors provided as the MAXREFDES117 development board were connected to the relevant SDA (blue wire) and SCL (green wire) pins on the I^2^C multiplexer from 0 to 7. A common 3.3 V power (red wire) and ground (black wire) for the I^2^C multiplexer and each MAX30102 sensor were connected to the 3.3 V and ground (GND) pins, respectively, on the NodeMCU microcontroller, with capacitors inline (not shown) to smooth the power supply.

**Figure 2 sensors-19-01874-f002:**
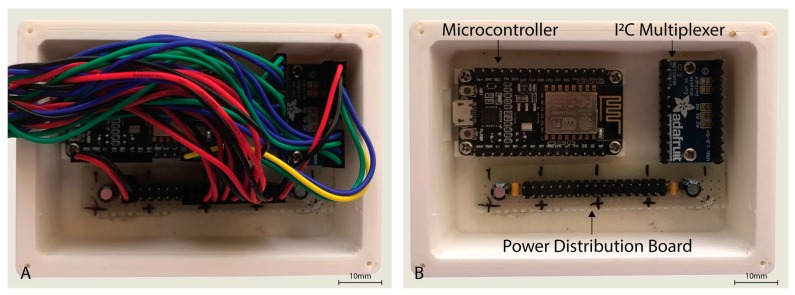
Housing for the NodeMCU microcontroller, I^2^C multiplexer, and power distribution board with cover removed (**A**,**B**) with wiring also removed.

**Figure 3 sensors-19-01874-f003:**
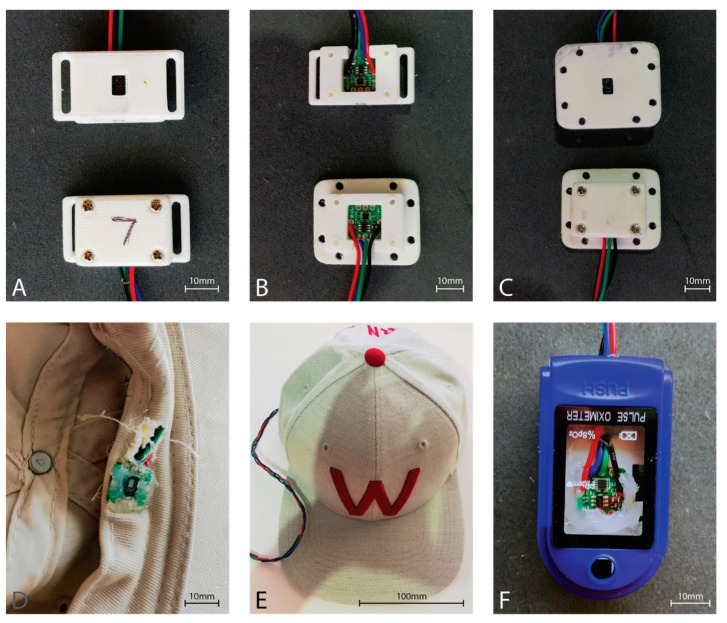
MAX30102 PPG sensor housings. (**A**) Strap type sensor housing used with a Velcro strip for attachment to a sports brace to the tibia. Top showing the skin-side, and bottom showing the outer housing. (**B**) Top shows the mounting of the sensor in the strap housing, while the bottom shows the sensor mounted in the housing used with sports tape. (**C**) Housing used with sports tape for sticking to the skin. Top showing the skin-side of the sensor housing, while the bottom shows the outer sensor housing. (**D**) Inside front brim of the hat showing the sensor board integrated into the brim. Note the neutral cure silicon protecting the sensor from perspiration during use. Wiring is fed behind the brim and exits behind the head. (**E**) The outside view of the hat with the sensor wire extending from the rear opening of the hat. (**F**) Modified commercial fingertip pulse oximeter housing. The hardware for the commercial pulse oximeter was removed from the housing, and the MAX30102 sensor board was affixed with neutral cure silicone such that the sensor window exited through the hole originally utilized by the commercial sensor.

**Figure 4 sensors-19-01874-f004:**
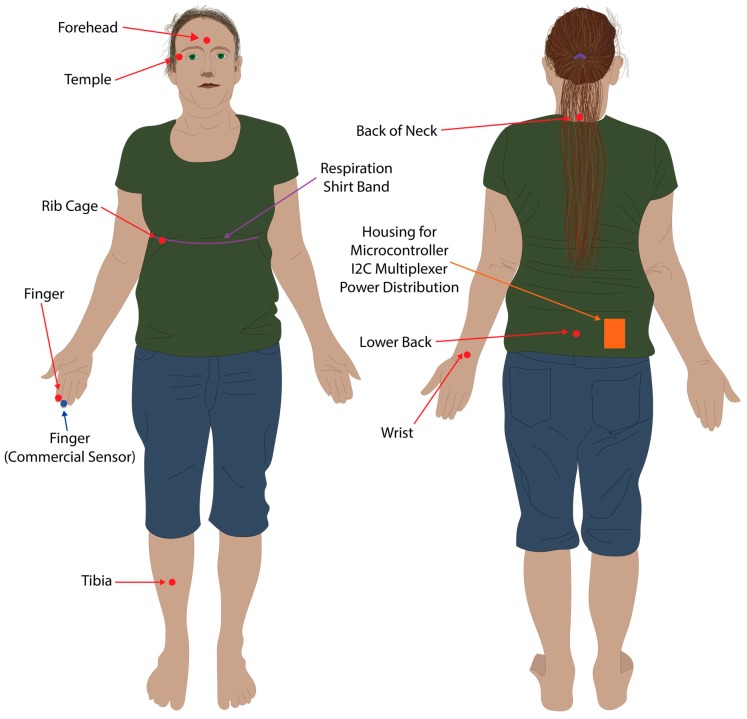
Sensor placement locations. Red dots and arrows indicate locations where the eight PPG sensors were placed. The blue dot and arrow indicates where the commercial pulse oximeter was located. The purple line and arrow indicates where the resistive band inside the respiration shirt was located for the respiration baseline data.

**Figure 5 sensors-19-01874-f005:**
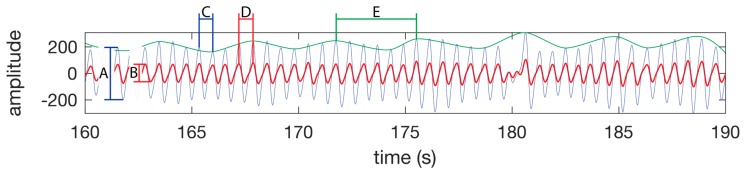
PPG waveform features. SpO_2_ is calculated using the amplitude of the IR waveform (A) and the amplitude of the red waveform (B) as per the Equation (1). (C) A single heart beat in the IR waveform. (D) A single heart beat in the red waveform. The time between peaks of a heat beat on a single channel is used to calculate the instantaneous heart rate. (E) A single respiration as seen as a change in amplitude of the IR or red waveform. The time between the peaks is used to calculate the instantaneous respiration rate. The green line is not part of the PPG waveform, but is presented in this figure to highlight the respiration component of the PPG waveform. The waveform was taken from the test subject 1 dataset while walking.

**Figure 6 sensors-19-01874-f006:**
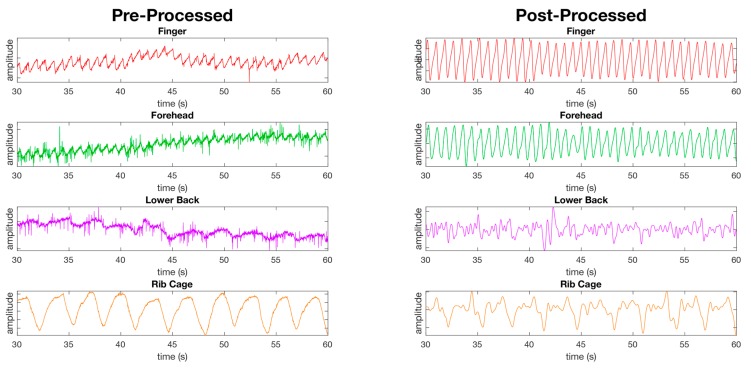
Figure showing the pre-processed waveforms (left) and the post-processed waveforms (right) from PPG sensors at four of the eight measurement locations using the red channel only at rest. The first two waveforms, the finger and forehead, show the success of the filters applied as detailed in the material and methods section revealing clear peaks and valleys in the waveforms. The remaining two waveforms (lower back and ribcage) show locations where the peaks and valleys are less defined post processing compared to the finger and forehead.

**Figure 7 sensors-19-01874-f007:**
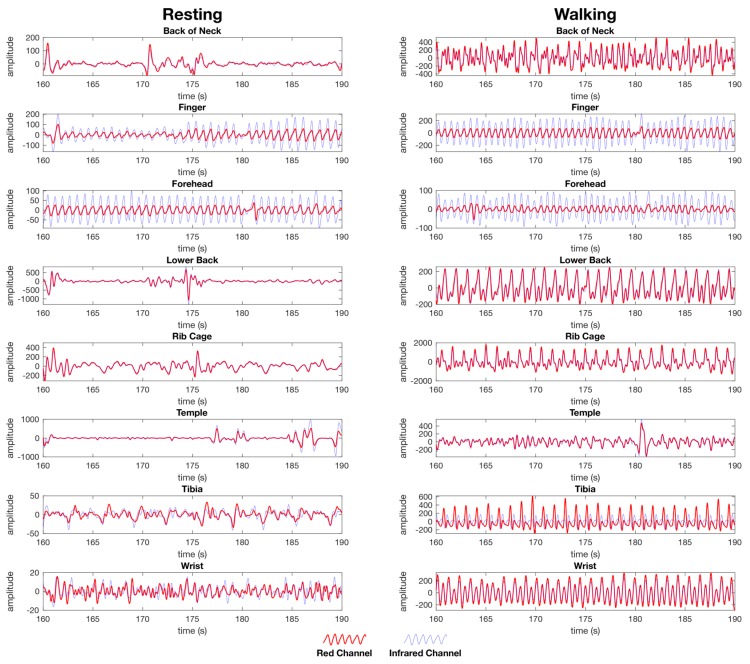
Post processed PPG waveforms used for heart rate and SpO_2_ calculation from the female dataset showing the red channel waveform (red) and the infrared channel waveform (blue) for both the resting (left) and walking (right) tests. The data is presented as the same randomly selected 30 second window of data from the same dataset for each location. Each anatomical location where the PPG sensors were simultaneously placed are represented, back of the neck, finger, forehead, lower back, rib cage, temple, tibia, and wrist. The y-axis units are arbitrary as the calculation of heart rate is performed on a time between peaks, whereas SpO_2_ is calculated as a ratio between the red and IR channels as set out in the materials and methods section.

**Figure 8 sensors-19-01874-f008:**
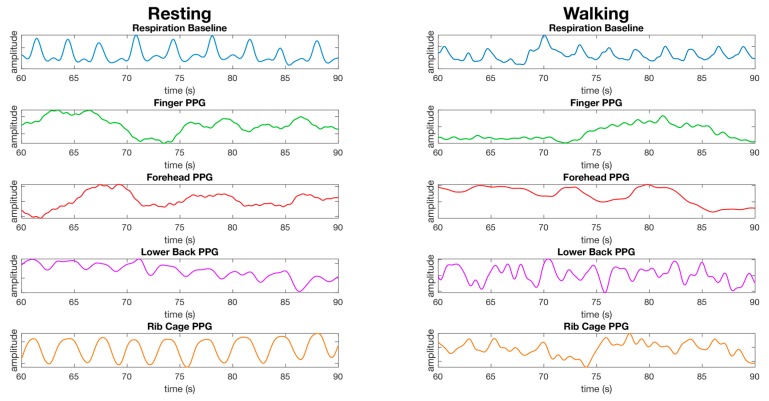
Comparison of the processed waveforms used for calculating respiration from the finger, forehead, lower back, and rib cage from a male dataset while resting (left) and while walking (right). The data is presented as the same randomly selected 30 s window of data from the same dataset for all resting waveforms, and the same dataset for walking waveforms. No synchronization of waveforms has been performed, the waveforms are presented as recorded bar processing as described in the material and methods section. The y-axis units are arbitrary as we are not comparing the amplitude of waveforms, but the average time between peaks over a window of data.

**Figure 9 sensors-19-01874-f009:**
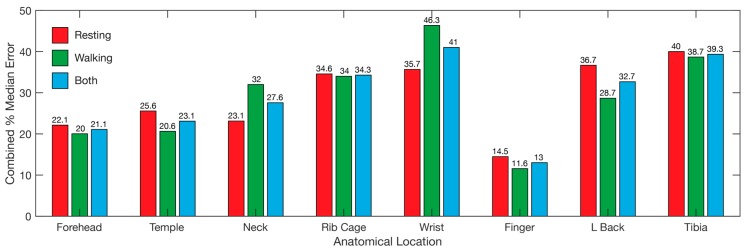
Overall error for each of the tested locations. Bars represent the median error for all three tested parameters, heart rate, SpO_2_, and respiration. Median error for resting (red), walking (blue), and both combined (green) are represented. Wrist refers to the posterior distal forearm. Neck refers to the back of the neck. L back refers to lower back.

**Table 1 sensors-19-01874-t001:** Measurement locations used by Nilsson et al. [18] showing the relationship between location and ability to measure heart rate and respiration.

Location	Heart Rate	Respiration
Forearm	Poor	Excellent
Finger (Reflective PPG)	Poor	Excellent
Finger (Transmissive PPG)	Poor	Excellent
Forehead	Moderate	Good
Wrist	Moderate	Good
Shoulder (560 nm)	Good	Good
Shoulder (806 nm)	Good	Good

**Table 2 sensors-19-01874-t002:** MAX30102 sensor configuration settings as set in the NodeMCU microcontroller firmware.

Setting	Value
Sampling Average	1 (no averaging)
FIFO Rolls on Full	Enabled
FIFO Almost Full Value	24
Mode Control	SpO_2_ Mode (Red and IR)
SpO_2_ ADC Range Control	4096
SpO_2_ Sample Rate Control	100 Samples per Second
LED Pulse Width Control	411 µs (18 bits)

**Table 3 sensors-19-01874-t003:** Results for heart rate calculation at eight anatomical locations presented as median percentage of error from the expected value as calculated from the commercial pulse oximeter recordings. Data recorded is an average error using 60 s windows of the instantaneous heart rate data for the red, infrared and combined (red and IR) channels. Wrist refers to the posterior distal forearm. Neck refers to the back of the neck. L Back refers to lower back. Results are rounded to two significant figures.

Location	Resting Median Error % (s.d.)			Walking Median Error % (s.d.)		
	Red	IR	Red & IR	Red	IR	Red and IR
Forehead	1.5 (±17)	0.68 (±1.5)	1.4 (±8.3)	4.6 (±7.8)	4.2 (±7.5)	4.3 (±7.6)
Temple	58 (±14)	5.9 (±23)	36 (±14)	8.3 (±17)	9.3 (±17)	9.2 (±17)
Neck	67 (±25)	31 (±36)	51 (±27)	11 (±28)	11 (±31)	13 (±28)
Rib Cage	56 (±18)	51 (±17)	52 (±13)	14 (±24)	14 (±25)	13 (±24)
Wrist	76 (±17)	62 (±32)	68 (±23)	10 (±10)	11 (±10)	11 (±10)
Finger	1.4 (±18)	1.2 (±16)	1.3 (±16)	6.8 (±11)	5.9 (±10)	6.5 (±10)
L Back	75 (±17)	76 (±15)	76 (±15)	19 (±25)	16 (±25)	18 (±25)
Tibia	83 (±12)	81 (±12)	81 (±11)	28 (±22)	34 (±20)	30 (±19)

**Table 4 sensors-19-01874-t004:** Results for SpO_2_ calculation at eight anatomical locations presented as a median percentage of error from the expected value as calculated from the commercial pulse oximeter. Data is recorded as an average error using 60 s windows of instantaneous SpO_2_ data. Wrist refers to the posterior distal forearm. Neck refers to the back of the neck. L Back refers to lower back. Results are rounded to two significant figures.

Location	Resting Median Error % (s.d.)	Walking Median Error (s.d.)
Forehead	2.0 (±1.1)	2.8 (±2.2)
Temple	2.7 (±7.7 × 10^4^)	6.7 (±5.9 × 10^2^)
Neck	8.4 (±47)	41 (±43)
Rib Cage	48 (±91)	54 (±52)
Wrist	20 (±29)	97 (±36)
Finger	2.1 (±1.2)	2.2 (±6.3)
L Back	22 (±2.4 × 10^3^)	19 (±1.6 × 10^3^)
Tibia	27 (±39)	28 (±2.7 × 10^2^)

**Table 5 sensors-19-01874-t005:** Median error for respiration calculation at eight anatomical location using the red and infrared channels at rest and walking. Median error is presented as the respirations per minute (rpm) deviation from the respiration shirt baseline data. Wrist refers to the posterior distal forearm. Neck refers to the back of the neck. L Back refers to lower back. Respirations per minute is denoted as rpm.

Location	Resting Median Error rpm (s.d.)		Walking Median Error rpm (s.d.)	
	Red	Infrared	Red	Infrared
Forehead	6.6 (±16)	18 (±21)	16 (±11)	15 (±11)
Temple	4.5 (±10)	5.4 (±12)	7.2 (±11)	8.5 (±14)
Neck	0.26 (±8.7)	1.1 (±8.7)	7.1 (±12)	8.7 (±12)
Rib Cage	0.13 (±6.6)	0.19 (±6.8)	2.9 (±13)	0.36 (±13)
Wrist	1.1 (±7.7)	0.2 (±7.7)	11 (±12)	8.4 (±13)
Finger	0.55 (±13)	0.11 (±13)	6.5 (±9.2)	3.9 (±10)
L Back	0.73 (±6.7)	0.73 (±6.5)	14 (±17)	13 (±17)
Tibia	0.29 (±8.3)	0.21 (±8.4)	26 (±11)	19 (±12)

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
