# Peer review of "A Comparison of Reflective Photoplethysmography for Detection of Heart Rate, Blood Oxygen Saturation, and Respiration Rate at Various Anatomical Locations"

_sensors, 2019, doi:10.3390/s19081874_

Round 1
Reviewer 1 Report
1. The abstract is an important section of a paper. Its main goal is to present in a clear and concise way, the central idea of the work and its novelty. The authors must improve the abstract section in this sense. They must focus their explanations.
2. We suggest that the conclusion section must be more focused. The first paragraph is interesting but long a general for this kind of section. It would be necessary to improve a bit this section
Author Response
We would like to thank you for taking your time to review our submission. Below is our response to your queries on our manuscript.
The abstract is an important section of a paper. Its main goal is to present in a clear and concise way, the central idea of the work and its novelty. The authors must improve the abstract section in this sense. They must focus their explanations.
Thank you for your feedback on the abstract. We have re-written the abstract to highlight the novelty and central idea of our study.
We suggest that the conclusion section must be more focused. The first paragraph is interesting but long a general for this kind of section. It would be necessary to improve a bit this section
As per the publisher guidelines for Sensors, a separate Conclusion section is optional, and therefore we did not include a defined conclusion in our paper. However, the authors acknowledge that the inclusion of a conclusion would be beneficial to the paper, and we have added a short conclusion to the paper. We thank you for alerting us to this improvement in our paper.
Reviewer 2 Report
Nice research, fine paper.
Authors investigated possibility of application of the optic sensor for heart rate, SpO2 and respiratory signals out of sensors located at different body locations. Presented results should be confirmed using more trials, on more volunteers, however selection of body location , especially in trials including movements should avoid presence of muscles as they interfere with reflected or transmitted light.
One question in the paper shuld be addreesed. Authors were using the commercial SpO2 meter with manal notation of obtained results. Data were noted every 15 s and compared to those, obtained by continous measurement. Question is - how rapid SpO2 changes are in real and how the commercial product is calculating this value ? Can we addure no time shift between two values ? I'd rather suggest some higher class meted allowing observation of red and infrared reflections together with calculated SpO2 values in future study.
Revise paper style. I think there is Results section misstyped / missplaced ?
Author Response
p.p1 {margin: 0.0px 0.0px 0.0px 0.0px; font: 9.5px 'Open Sans'; color: #5488cf} p.p2 {margin: 0.0px 0.0px 0.0px 0.0px; font: 9.5px 'Open Sans'} p.p3 {margin: 0.0px 0.0px 0.0px 0.0px; font: 9.5px 'Open Sans'; min-height: 13.0px}We wish to thank you for taking time to review our submission. Below is our response to your queries regarding our manuscript.
Nice research, fine paper.
Thankyou for your kind feedback.
Authors investigated possibility of application of the optic sensor for heart rate, SpO2 and respiratory signals out of sensors located at different body locations. Presented results should be confirmed using more trials, on more volunteers, however selection of body location , especially in trials including movements should avoid presence of muscles as they interfere with reflected or transmitted light.
Thank you for your feedback on the design of our study. We decided upon the locations as we were investigating locations where a PPG sensor could be worn for an extended period of time, with minimal interference to day to day activities. Some of those locations, for example the tibia and wrist do have muscles, however in the case of the tibia we aimed for the part where the tissue is thinnest. As for the wrist, we measured the location where wrist worn devices are already used, such as the Apple Watch. Unfortunately these locations proved to be inappropriate for measurement of the selected parameters using PPG sensors, which enabled us to rule out those locations for our future study.
We do agree that a larger study group is needed to validate the results further, and we do plan on extending the study to include a larger set of volunteers in the future, as well as include the green wavelength, which appears to be less prone to motion artifacts, as well as red and IR.
One question in the paper shuld be addreesed. Authors were using the commercial SpO2 meter with manal notation of obtained results. Data were noted every 15 s and compared to those, obtained by continous measurement. Question is - how rapid SpO2 changes are in real and how the commercial product is calculating this value ? Can we addure no time shift between two values ? I'd rather suggest some higher class meted allowing observation of red and infrared reflections together with calculated SpO2 values in future study.
Unfortunately constraints to the budget at this stage of our overall project eliminated the ability for us to use an higher class and automated method of ground truth PPG data. We agree that it would be beneficial to use an FDA approved pulse oximeter where the waveforms for red and IR can be collected, along with the devices calculations for HR, SpO2 and respiration in real time, synchronized to our multi-headed PPG sensors. In future studies we intend to utilise a commercial pulse oximeter whereby we can access the raw infrared and red data in real time in synch with our sensors.
Revise paper style. I think there is Results section misstyped / missplaced ?
Thankyou for pointing this out, it was overlooked by our team. The section heading was misplaced on the line above where it should have been. This has been resolved.s